# A Smoother Way to Train Structured Prediction Models

**Krishna Pillutla,  Vincent Roulet,  Sham M. Kakade,  Zaid Harchaoui**

Paul G. Allen School of Computer Science & Engineering and Department of Statistics

University of Washington

name@uw.edu

## Abstract

We present a framework to train a structured prediction model by performing smoothing on the inference algorithm it builds upon. Smoothing overcomes the non-smoothness inherent to the maximum margin structured prediction objective, and paves the way for the use of fast primal gradient-based optimization algorithms. We illustrate the proposed framework by developing a novel primal incremental optimization algorithm for the structural support vector machine. The proposed algorithm blends an extrapolation scheme for acceleration and an adaptive smoothing scheme and builds upon the stochastic variance-reduced gradient algorithm. We establish its worst-case global complexity bound and study several practical variants. We present experimental results on two real-world problems, namely named entity recognition and visual object localization. The experimental results show that the proposed framework allows us to build upon efficient inference algorithms to develop large-scale optimization algorithms for structured prediction which can achieve competitive performance on the two real-world problems.

## 1   Introduction

Consider the optimization problem arising when training structural support vector machines:

$$\min_{\boldsymbol{w}\in\mathbb{R}^d}\left[F(\boldsymbol{w}) := \frac{1}{n}\sum_{i=1}^{n}f^{(i)}(\boldsymbol{w}) + \frac{\lambda}{2}\|\boldsymbol{w}\|_2^2\right],\tag{1}$$

where each $f^{(i)}$ is the structural hinge loss. Structural support vector machines were designed for prediction problems where outputs are discrete data structures such as sequences or trees [59, 65].

Batch nonsmooth optimization algorithms such as cutting plane methods are appropriate for problems with small or moderate sample sizes [65, 21]. Stochastic nonsmooth optimization algorithms such as stochastic subgradient methods can tackle problems with large sample sizes [49, 57]. However both families of methods achieve the typical worst-case complexity bounds of nonsmooth optimization algorithms and cannot easily leverage a possible hidden smoothness of the objective.

Furthermore, as significant progress is being made on incremental smooth optimization algorithms for training unstructured prediction models [36], we would like to transfer such advances and design faster optimization algorithms to train structured prediction models. Indeed if each term in the finite-sum were $L$-smooth [1], incremental optimization algorithms such as MISO [37], SAG [33, 53], SAGA [10], SDCA [55], and SVRG [23] could leverage the finite-sum structure of the objective (1) and achieve faster convergence than batch algorithms on large-scale problems.

Incremental optimization algorithms can be further accelerated, either on a case-by-case basis [56, 14, 1, 9] or using the Catalyst acceleration scheme [35, 36], to achieve near-optimal convergence rates [67]. Accelerated incremental optimization algorithms demonstrate stable and fast convergence behavior on a wide range of problems, in particular for ill-conditioned ones.

We introduce a general framework that allows us to bring the power of accelerated incremental optimization algorithms to the realm of structured prediction problems. To illustrate our framework, we focus on the problem of training a structural support vector machine (SSVM). The same ideas can be applied to other structured prediction models to obtain faster training algorithms.

We seek primal optimization algorithms, as opposed to saddle-point or primal-dual optimization algorithms, in order to be able to tackle structured prediction models with affine mappings such as SSVM as well as deep structured prediction models with nonlinear mappings. We show how to shade off the inherent non-smoothness of the objective while still being able to rely on efficient inference algorithms.

**Smooth inference oracles.** We introduce a notion of smooth inference oracles that gracefully fits the framework of black-box first-order optimization. While the exp inference oracle reveals the relationship between max-margin and probabilistic structured prediction models, the top-$K$ inference oracle can be efficiently computed using simple modifications of efficient inference algorithms in many cases of interest.

**Incremental optimization algorithms.** We present a new algorithm built on top of SVRG, blending an extrapolation scheme for acceleration and an adaptive smoothing scheme. We establish the worst-case complexity bounds of the proposed algorithm and demonstrate its effectiveness compared to competing algorithms on two tasks, namely named entity recognition and visual object localization.

The code is publicly available on the authors' websites. All the proofs are provided in [48].

## 2  Smoothing Inference for Structured Prediction

Given an input $x \in \mathcal{X}$ of arbitrary structure, e.g. a sentence, a structured prediction model outputs its prediction as a structured object $y \in \mathcal{Y}$, such as a parse tree, where the set of all outputs $\mathcal{Y}$ may be finite yet often large. The score function $\phi$, parameterized by $w \in \mathbb{R}^d$, quantifies the compatibility of an input $x$ and an output $y$ as $\phi(x, y; w)$. It is assumed to decompose onto the structure at hand such that the inference problem $y^*(x; w) \in \mathrm{argmax}_{y \in \mathcal{Y}} \phi(x, y; w)$ can be solved efficiently by a combinatorial optimization algorithm. Training a structured prediction model then amounts to finding the best score function such that the inference procedure provides correct predictions.

**Structural hinge loss.** The standard formulation uses a feature map $\Phi : \mathcal{X} \times \mathcal{Y} \to \mathbb{R}^d$ such that score functions are linear in $w$, i.e. $\phi(x, y; w) = \Phi(x, y)^\top w$. The structural hinge loss, an extension of binary and multi-class hinge losses, considers a majorizing surrogate of a given loss function $\ell$ such as the Hamming loss, that measures the error incurred by predicting $y^*(x; w)$ on a sample $(x, y)$ as $\ell(y, y^*(x; w))$. For an input-output pair $(x_i, y_i)$, the structural hinge loss is defined as

$$f^{(i)}(w) = \max_{y' \in \mathcal{Y}} \left\{ \phi(x_i, y'; w) + \ell(y_i, y') \right\} - \phi(x_i, y_i; w) = \max_{y' \in \mathcal{Y}} \psi_i(y'; w), \qquad (2)$$

where $\psi_i(y'; w) := \phi(x_i, y'; w) + \ell(y_i, y') - \phi(x_i, y_i; w) = a_{i,y'}^\top w + b_{i,y'}$ is the augmented score function, an affine function of $w$. The loss $\ell$ is also assumed to decompose onto the structure so that the maximization in (2), also known as loss augmented inference, is no harder than the inference problem consisting in computing $y^*(x; w)$. The learning problem (1) is the minimization of the structural hinge losses on the training data $(x_i, y_i)_{i=1}^n$ with a regularization penalty. We shall refer to a generic term $f(w) = \max_{y' \in \mathcal{Y}} \psi(y'; w)$ in the finite-sum from now on.

**Smoothing strategy.** To smooth the structural hinge loss, we decompose it as the composition of the max function with a linear mapping. The former can then be easily smoothed through its dual formulation to obtain a smooth surrogate of (2). Formally, define the mapping $g$ and the max function $h$ respectively as

$$g : \begin{cases} \mathbb{R}^d & \to \mathbb{R}^m \\ w & \mapsto (\psi(y'; w))_{y' \in \mathcal{Y}} = Aw + b \end{cases}, \qquad h : \begin{cases} \mathbb{R}^m & \to \mathbb{R} \\ z & \mapsto \max_{i \in [m]} z_i \end{cases}, \qquad (3)$$

where $m = |\mathcal{Y}|$. The structural hinge loss can now be expressed as $f = h \circ g$.

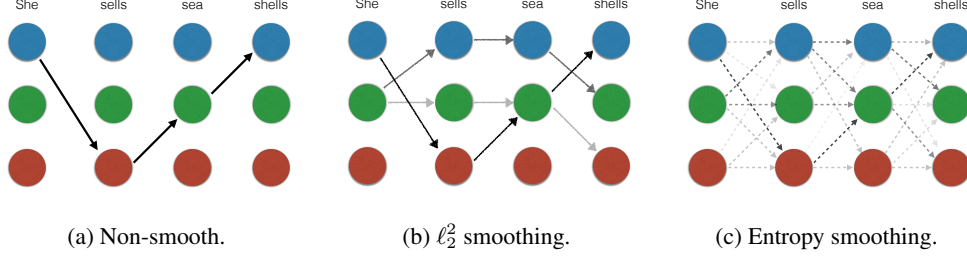

(a) Non-smooth.       (b) $\ell_2^2$ smoothing.       (c) Entropy smoothing.

Figure 1: Viterbi trellis for a chain graph with four nodes and three labels.

The max function can be written as $h(\boldsymbol{z}) = \max_{i \in [m]} z_i = \max_{\boldsymbol{u} \in \Delta^{m-1}} \boldsymbol{z}^\top \boldsymbol{u}$ where $\Delta^{m-1}$ is the probability simplex in $\mathbb{R}^m$. Its simplicity allows us to analytically compute its infimal convolution with a smooth function [2]. The smoothing $h_{\mu\omega}$ of $h$ by a strongly convex function $\omega$ with smoothing coefficient $\mu > 0$ is defined as

$$h_{\mu\omega}(\boldsymbol{z}) := \max_{\boldsymbol{u} \in \Delta^{m-1}} \left\{ \boldsymbol{z}^\top \boldsymbol{u} - \mu\omega(\boldsymbol{u}) \right\},$$

whose gradient is the maximizer of the above expression. The smooth approximation of the structural hinge loss is then given by $f_{\mu\omega} := h_{\mu\omega} \circ \boldsymbol{g}$. This smoothing technique was introduced by Nesterov [43] who showed that if $\omega$ is 1-strongly convex with respect to $\| \cdot \|_\alpha$, then $f_{\mu\omega}$ is $(\|\boldsymbol{A}\|_{2,\alpha}^2/\mu)$-smooth[2], and approximates $f$ for any $\boldsymbol{w}$ as

$$\mu \min_{\boldsymbol{u} \in \Delta^{m-1}} \omega(\boldsymbol{u}) \leq f(\boldsymbol{w}) - f_{\mu\omega}(\boldsymbol{w}) \leq \mu \max_{\boldsymbol{u} \in \Delta^{m-1}} \omega(\boldsymbol{u}).$$

**Smoothing variants.** We focus on the negative entropy and the squared Euclidean norm as choices for $\omega$, denoted respectively

$$-H(\boldsymbol{u}) := \sum_{i=1}^m u_i \log u_i \qquad \text{and} \qquad \ell_2^2(\boldsymbol{u}) := \tfrac{1}{2}(\|\boldsymbol{u}\|_2^2 - 1).$$

The gradient of their corresponding smooth counterparts can be computed respectively by the softmax and the orthogonal projection onto the simplex, i.e.

$$\nabla h_{-\mu H}(\boldsymbol{z}) = \left[ \frac{\exp(z_i/\mu)}{\sum_{j=1}^m \exp(z_j/\mu)} \right]_{i=1,\ldots,m} \qquad \text{and} \qquad \nabla h_{\mu \ell_2^2}(\boldsymbol{z}) = \operatorname{proj}_{\Delta^{m-1}}(\boldsymbol{z}/\mu).$$

The gradient of the smooth surrogate $f_{\mu\omega}$ can be written using the chain rule. This involves computing $\nabla \boldsymbol{g}$ along all $m = |\mathcal{Y}|$ of its components, which may be intractable. However, for the $\ell_2^2$ smoothing, the gradient $\nabla h_{\mu \ell_2^2}(\boldsymbol{z})$ is given by the projection of $\boldsymbol{z}/\mu$ onto the simplex, which selects a small number, denoted $K_{\boldsymbol{z}/\mu}$, of its largest coordinates. We shall approximate this projection by fixing $K$ independently of $\boldsymbol{z}/\mu$ and defining

$$h_{\mu,K}(\boldsymbol{z}) = \max_{\boldsymbol{u} \in \Delta^{K-1}} \left\{ \boldsymbol{z}_{[K]}^\top \boldsymbol{u} - \mu \ell_2^2(\boldsymbol{u}) \right\},$$

as an approximation of $h_{\mu \ell_2^2}(\boldsymbol{z})$, where $\boldsymbol{z}_{[K]} \in \mathbb{R}^K$ denote the $K$ largest components of $\boldsymbol{z}$. If $K_{\boldsymbol{z}/\mu} < K$ this approximation is exact and for fixed $\boldsymbol{z}$, this holds for small enough $\mu$, as shown in [48]. The resulting surrogate is denoted $f_{\mu,K} = h_{\mu,K} \circ \boldsymbol{g}$.

**Smooth inference oracles.** We define a smooth inference oracle as a first-order oracle for a smooth counterpart of the structural hinge loss. Recall that a first-order oracle for a function $f$ is a numerical routine which, given a point $\boldsymbol{w} \in \operatorname{dom}(f)$, returns the function value $f(\boldsymbol{w})$ and a (sub)gradient $v \in \partial f(\boldsymbol{w})$. We define three variants of a smooth inference oracle: i) the max oracle; ii) the exp oracle; iii) the top-$K$ oracle. The max oracle corresponds to the usual inference oracle in maximum margin structured prediction, while the exp oracle and the the top-$K$ oracle correspond resp. to the entropy-based and $\ell_2^2$-based smoothing.

Figure 1 illustrates the notion on a chain structured output. The inference problem is non-smooth and a small change in $\boldsymbol{w}$ might lead to a radical change in the best scoring path as shown in Fig. 1a. The $\ell_2^2$-based smooth inference amounts to picking some number of the top scoring paths. Notice the sparsity pattern in Fig. 1b. The entropy-based smooth inference amounts to weighting all paths, with a higher weight for top scoring paths as shown in Fig. 1c.

Table 1: Smooth inference oracles, algorithms and complexity. Here, $p$ is the size of each $\boldsymbol{y} \in \mathcal{Y}$. The time complexity is phrased in terms of the time complexity $\mathcal{T}$ of the max oracle.

| Max oracle | Top-$K$ oracle | | Exp oracle | |
| Algo | Algo | Time | Algo | Time |
|---|---|---|---|---|
| Dynamic Programming | Top-$K$ DP | $O(K\mathcal{T}\log K)$ | Sum-Product | $O(\mathcal{T})$ |
| Graph cut | BMMF | $O(pK\mathcal{T})$ | Intractable | |
| Graph matching | BMMF | $O(K\mathcal{T})$ | Intractable | |
| Branch and Bound search | Top-$K$ search | N/A | Intractable | |

**Definition 1.** Consider $f(\boldsymbol{w}) = \max_{\boldsymbol{y}' \in \mathcal{Y}} \psi(\boldsymbol{y}'; \boldsymbol{w})$ and $\boldsymbol{w} \in \mathbb{R}^d$,

- the *max oracle* returns $f(\boldsymbol{w})$ and $\nabla\psi(\boldsymbol{y}^*; \boldsymbol{w}) \in \partial f(\boldsymbol{w})$, where $\boldsymbol{y}^* \in \operatorname{argmax}_{\boldsymbol{y}' \in \mathcal{Y}} \psi(\boldsymbol{y}'; \boldsymbol{w})$;

- the *exp oracle* returns $f_{-\mu H}(\boldsymbol{w})$ and $\nabla f_{-\mu H}(\boldsymbol{w}) = \mathbb{E}_{\boldsymbol{y}' \sim p_\mu}[\nabla\psi(\boldsymbol{y}'; \boldsymbol{w})]$, where $p_\mu(\boldsymbol{y}') \propto \exp(\psi(\boldsymbol{y}'; \boldsymbol{w})/\mu)$;

- the *top-$K$ oracle* computes the $K$ best outputs $\{\boldsymbol{y}_{(i)}\}_{i=1}^K = \mathcal{Y}_K$ satisfying

$$\psi(\boldsymbol{y}_{(1)}; \boldsymbol{w}) \geq \cdots \geq \psi(\boldsymbol{y}_{(K)}; \boldsymbol{w}) \geq \max_{\boldsymbol{y}' \in \mathcal{Y}\backslash\mathcal{Y}_K} \psi(\boldsymbol{y}'; \boldsymbol{w})$$

to return $f_{\mu,K}(\boldsymbol{w})$ and $\nabla f_{\mu,K}(\boldsymbol{w})$ as surrogates for $f_{\mu\ell_2^2}(\boldsymbol{w})$ and $\nabla f_{\mu\ell_2^2}(\boldsymbol{w})$.

On the one hand, the entropy-based smoothing of a structural support vector machine somewhat interpolates between a regular structural support vector machine and a conditional random field [31] through the smoothing parameter $\mu$. On the other hand, the $\ell_2^2$-based smoothing only requires a top-$K$ oracle, making it a more practical option, as illustrated in Table 1.

**Smooth inference algorithms.** The implementation of inference oracles depends on the structure of the output, given by a probabilistic graphical model [48]. When the latter is a tree, exact procedures are available, otherwise some algorithms may not be practical. See Table 1 for a summary[3]. The formal description, algorithms and proofs of correctness are provided in [48].

*Dynamic Programming.* For graphs with a tree structure or bounded tree-width, the *max oracle* is implemented by dynamic programming (DP) algorithms such as the popular Viterbi algorithm. The *exp-oracle* can be achieved by replacing the max in DP with log-sum-exp and using back-propagation at $O(1)$ times the cost of the max oracle. The *top-$K$ oracle* is implemented by the top-$K$ DP algorithm which keeps track of the $K$ largest intermediate scores and the back-pointers at $O(K)$ times the cost of the max oracle; see [48] for details.

*Graph cut and matching.* For specific probabilistic graphical models, exact inference is possible in loopy graphs by the use of graph cuts [27] or perfect matchings in bipartite graphs [60]. In this case, a top-$K$ oracle can be implemented by the best max marginal first (BMMF) algorithm [68] at $2K$ computations of max-marginals, which can be efficiently computed for graph cuts [26] and matchings [12]. The exp oracle is intractable (in fact, it is #P-complete) [20].

*Branch and bound search.* In special cases, branch and bound search allows exact inference in loopy graphs by partitioning $\mathcal{Y}$ and exploring promising parts first using a heuristic. Examples include the celebrated efficient subwindow search [32] in computer vision or A$^\star$ algorithm in natural language processing [34, 18]. Here, the top-$K$ oracle can be implemented by letting the search run until $K$ outputs are found while the exp oracle is intractable. The running time of both the max and top-$K$ oracles depends on the heuristic used and might be exponential in the worst case.

**Algorithm 1** Catalyst with smoothing
---

1: **Input:** Objective $F$ in (1), linearly convergent method $\mathcal{M}$, initial $\boldsymbol{w}_0$, $\alpha_0 \in (0,1)$.
   Smoothing $(\mu_k)_{k\geq 1}$ and regularization $(\kappa_k)_{k\geq 1}$ parameters, relative accuracies $(\delta_k)_{k\geq 1}$.
2: **Initialize:** $\boldsymbol{z}_0 = \boldsymbol{w}_0$.
3: **for** $k = 1$ **to** $T$ **do**
4:     Using $\mathcal{M}$ with $\boldsymbol{z}_{k-1}$ as the starting point, find

$$\boldsymbol{w}_k \approx \underset{\boldsymbol{w} \in \mathbb{R}^d}{\operatorname{argmin}} \left[ F_{\mu_k\omega,\kappa_k}(\boldsymbol{w};\boldsymbol{z}_{k-1}) := \frac{1}{n}\sum_{i=1}^n f^{(i)}_{\mu_k\omega}(\boldsymbol{w}) + \frac{\lambda}{2}\|\boldsymbol{w}\|_2^2 + \frac{\kappa_k}{2}\|\boldsymbol{w} - \boldsymbol{z}_{k-1}\|_2^2 \right], \tag{5}$$

   such that $F_{\mu_k\omega,\kappa_k}(\boldsymbol{w}_k;\boldsymbol{z}_{k-1}) - \min_{\boldsymbol{w}} F_{\mu_k\omega,\kappa_k}(\boldsymbol{w};\boldsymbol{z}_{k-1}) \leq \frac{\delta_k\kappa_k}{2}\|\boldsymbol{w}_k - \boldsymbol{z}_{k-1}\|_2^2$.
5:     Compute $\alpha_k$ and $\beta_k$ such that

$$\alpha_k^2(\kappa_{k+1} + \lambda) = (1-\alpha_k)\alpha_{k-1}^2(\kappa_k + \lambda) + \alpha_k\lambda, \qquad \beta_k = \frac{\alpha_{k-1}(1-\alpha_{k-1})(\kappa_k+\lambda)}{\alpha_{k-1}^2(\kappa_k+\lambda)+\alpha_k(\kappa_{k+1}+\lambda)}.$$

6:     Set $\boldsymbol{z}_k = \boldsymbol{w}_k + \beta_k(\boldsymbol{w}_k - \boldsymbol{w}_{k-1})$.
7: **end for**
8: **return** $\boldsymbol{w}_T$.
---

## 3   Catalyst with smoothing

For a single input-output pair ($n=1$), the problem (1) is $\min_{\boldsymbol{w}\in\mathbb{R}^d} h(\boldsymbol{A}\boldsymbol{w}+\boldsymbol{b}) + \frac{\lambda}{2}\|\boldsymbol{w}\|_2^2$, where $h$ is a simple non-smooth convex function. The Nesterov smoothing technique overcomes the non-smoothness of the objective by considering a smooth surrogate instead [43, 42]. We combine this with the Catalyst scheme to accelerate a linearly-convergent smooth optimization algorithm [36].

**Catalyst with smoothing.** The Catalyst approach considers at each outer iteration a regularized objective centered around the current iterate [36]. The algorithm proceeds by performing approximate proximal point steps, that is from a point $\boldsymbol{z}$ and for a step-size $1/\kappa$ one computes the minimizer of $\min_{\boldsymbol{w}\in\mathbb{R}^m} F(\boldsymbol{w}) + \frac{\kappa}{2}\|\boldsymbol{w}-\boldsymbol{z}\|_2^2$. We only need an approximate solution returned by a given optimization method $\mathcal{M}$ that enjoys a linear convergence guarantee.

We extend the Catalyst approach to non-smooth optimization problems by performing adaptive smoothing in the outer-loop and adjusting the level of accuracy accordingly in the inner-loop. We define

$$F_{\mu\omega,\kappa}(\boldsymbol{w};\boldsymbol{z}) := \frac{1}{n}\sum_{i=1}^n f^{(i)}_{\mu_k\omega}(\boldsymbol{w}) + \frac{\lambda}{2}\|\boldsymbol{w}\|_2^2 + \frac{\kappa}{2}\|\boldsymbol{w}-\boldsymbol{z}\|_2^2 \tag{4}$$

as a smooth surrogate to the objective centered around a given point $\boldsymbol{z} \in \mathbb{R}^d$. Note that the original Catalyst considered a fixed regularization term $\kappa$ [36], while we vary $\kappa$ and $\mu$. Doing so enables us to get adaptive smoothing strategies.

The proposed inner-outer scheme is presented in Algorithm 1. In view of the strong convexity of $F_{\mu_k\omega,\kappa_k}(\cdot;\boldsymbol{z}_{k-1})$, the stopping criterion for the subproblem (5) can be checked by looking at the gradient of $F_{\mu_k\omega,\kappa_k}(\cdot;\boldsymbol{z}_{k-1})$. As it is smooth and strongly convex, the maximal number of iterations to satisfy the stopping criterion can also be derived. In practice, however, we recommend a practical variant similar in spirit to the one proposed by [36] that lets $\mathcal{M}$ run for a fixed budget of iterations in each inner loop. Below, we denote $\boldsymbol{w}^* \in \operatorname{argmin}_{\boldsymbol{w}\in\mathbb{R}^d} F(\boldsymbol{w})$ and $F^* = F(\boldsymbol{w}^*)$.

**Theorem 1.** *Consider problem (1) and a smoothing function $\omega$ s.t. $-D \leq \omega(\boldsymbol{u}) \leq 0$ for all $\boldsymbol{u} \in \Delta$. Assume parameters $(\mu_k)_{k\geq 1}$, $(\kappa_k)_{k\geq 1}$, $(\delta_k)_{k\geq 1}$ of Algorithm 1 are non-negative with $(\mu_k)_{k\geq 1}$ non-increasing, $\delta_k \in [0,1)$, and $\alpha_k \in (0,1)$ for all k. Then, Algorithm 1 generates $(\boldsymbol{w}_k)_{k\geq 0}$ such that*

$$F(\boldsymbol{w}_k) - F^* \leq \frac{\mathcal{A}_0^{k-1}}{\mathcal{B}_1^k}\Delta_0 + \mu_k D + \sum_{j=1}^k \frac{\mathcal{A}_j^{k-1}}{\mathcal{B}_j^k}(\mu_{j-1} - (1-\delta_j)\mu_j)D, \tag{6}$$

*where $\mathcal{A}_j^k = \prod_{i=j}^k(1-\alpha_i)$, $\mathcal{B}_j^k = \prod_{i=j}^k(1-\delta_i)$, $\Delta_0 = F(\boldsymbol{w}_0) - F^* + \frac{(\kappa_1+\lambda)\alpha_0^2 - \lambda\alpha_0}{2(1-\alpha_0)}\|\boldsymbol{w}_0 - \boldsymbol{w}^*\|_2^2$, and $\mu_0 = 2\mu_1$.*

Table 2: Summary of global complexity of SC-SVRG, i.e., Algorithm 1 with SVRG as the inner solver for various parameter settings. We show $\mathbb{E}[N]$, the expected total number of SVRG iterations required to obtain an accuracy $\epsilon$, up to constants and factors logarithmic in problem parameters. We denote $\Delta F_0 := F(\boldsymbol{w}_0) - F^*$ and $\Delta_0 = \|\boldsymbol{w}_0 - \boldsymbol{w}^*\|_2$. Constants $a$ and $D$ are defined so that $-D \leq \omega(\boldsymbol{u}) \leq 0$ for all $\boldsymbol{u} \in \Delta$ and each $f_{\mu\omega}^{(i)}$ is $a/\mu$-smooth for $i \in [n]$.

| Scheme | $\lambda > 0$ | $\mu_k$ | $\kappa_k$ | $\delta_k$ | $\mathbb{E}[N]$ | Remark |
|---|---|---|---|---|---|---|
| 1 | Yes | $\frac{\epsilon}{D}$ | $\frac{aD}{\epsilon n} - \lambda$ | $\sqrt{\frac{\lambda \epsilon n}{aD}}$ | $n + \sqrt{\frac{aDn}{\lambda \epsilon}}$ | fix $\epsilon$ in advance |
| 2 | Yes | $\mu c^k$ | $\lambda$ | $c'$ | $n + \frac{a}{\lambda \epsilon} \frac{\Delta F_0 + \mu D}{\mu}$ | $c, c' < 1$ are universal constants |
| 3 | No | $\epsilon/D$ | $aD/\epsilon n$ | $1/k^2$ | $n\sqrt{\frac{\Delta F_0}{\epsilon}} + \frac{\sqrt{aDn}\Delta_0}{\epsilon}$ | fix $\epsilon$ in advance |
| 4 | No | $\mu/k$ | $\kappa_0 k$ | $1/k^2$ | $\frac{\widehat{\Delta}_0}{\epsilon}\left(n + \frac{a}{\mu\kappa_0}\right)$ | $\widehat{\Delta}_0 = \Delta F_0 + \frac{\kappa_0}{2}\Delta_0^2 + \mu D$ |

Theorem 1 establishes the complexity of the Catalyst smoothing scheme for a general smoothing function and a general linearly-convergent smooth optimization algorithm $\mathcal{M}$.

Using Theorem 1, we can derive strategies for strongly or non-strongly convex objectives ($\lambda > 0$ or not) with adaptive smoothing that vanishes over time to get progressively better surrogates of the original objective.

The global complexity of the algorithm depends then on the choice of $\mathcal{M}$. We present in Table 2 the total complexity for different strategies when SVRG [23] is used as $\mathcal{M}$, resulting in an algorithm called SC-SVRG in the remainder of the paper. Note that the adaptive smoothing schemes (2, 4) do not match the rate obtained by a fixed smoothing (1, 3). A standard doubling trick can easily fix this. Yet we choose to use an adaptive smoothing scheme, easier to use and working well in practice (see Sec. 4). All proofs are given in [48].

**Extension to nonlinear mappings.** When the score function is not linear in $\boldsymbol{w}$, the overall problem is not convex in general. However, if the score function is smooth, then one could take advantage of the composite structure of the structural hinge loss $f = h \circ \boldsymbol{g}$ by using the prox-linear algorithm [6, 11]. At each step, the latter linearizes the mapping $\boldsymbol{g}$ around the current iterate $\boldsymbol{w}_k$, resulting in a convex model $\boldsymbol{w} \mapsto h(\boldsymbol{w}_k + \nabla \boldsymbol{g}(\boldsymbol{w}_k)^\top(\boldsymbol{w} - \boldsymbol{w}_k))$ of $h \circ \boldsymbol{g}$ around $\boldsymbol{w}_k$. The overall convex model of the objective $F$ with an additional proximal term is then minimized. The next iterate is given by

$$\boldsymbol{w}_{k+1} = \operatorname*{argmin}_{\boldsymbol{w} \in \mathbb{R}^d} \frac{1}{n} \sum_{i=1}^{n} h(\boldsymbol{g}^{(i)}(\boldsymbol{w}) + \nabla \boldsymbol{g}^{(i)}(\boldsymbol{w}_k)^\top(\boldsymbol{w} - \boldsymbol{w}_k)) + \frac{\lambda}{2}\|\boldsymbol{w}\|_2^2 + \frac{1}{2\gamma}\|\boldsymbol{w} - \boldsymbol{w}_k\|^2 \quad (7)$$

where $\boldsymbol{g}^{(i)}$ is the mapping associated with the $i^{\text{th}}$ sample and $\gamma > 0$ is the parameter of the proximal term. This subproblem reduces to training a structured prediction model with an affine augmented score function. Therefore we can solve it with the SC-SVRG algorithm introduced earlier. Note that only approximate solutions are required to get a global convergence to a stationary point [11]. The theoretical analysis and numerical experiments showing the potential of this approach compared to subgradient methods can be found in [48].

## 4 Experiments

We compare the proposed algorithm and several competing algorithms to train a structural support vector machine on the tasks of named entity recognition and visual object localization. Additional details on the datasets, algorithms, parameters as well as an extensive evaluation in different settings can be found in [48]. We use the $\ell_2^2$-based smoothing in all experiments as explained in Sec. 2.

**Named entity Recognition.** The task consists in predicting the tagging of a sequence into named entities. We consider the CoNLL 2003 dataset with $n = 14987$ [63]. The Viterbi algorithm provides an efficient max oracle and the top-$K$ oracle is obtained following the discussion in Sec. 2. The loss $\ell$ is the Hamming loss here. The features $\Phi(\boldsymbol{x}, \boldsymbol{y})$ are obtained from the local context around each word [64]. We use the $F_1$ score as the performance metric for evaluation.

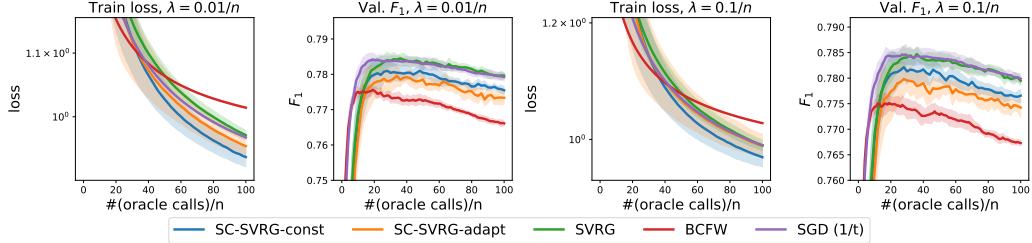

(a) Performance on CoNLL-2003 for named entity recognition.

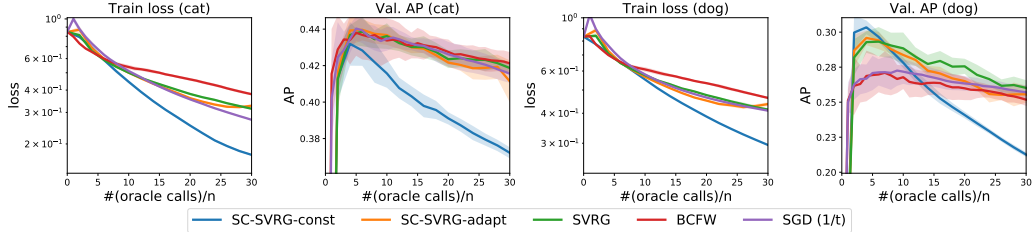

(b) Sample performance on PASCAL VOC 2007 for visual object localization for $\lambda = 10/n$.

Figure 2: Experimental comparison of proposed methods for the tasks of named entity recognition (Fig. 2a) and visual object localization (Fig. 2b). All shaded areas represent one standard deviation over ten random runs. See [48] for all plots.

**Visual object localization.** The task consists in predicting the spatial location of the visual object in an image. We consider the PASCAL VOC 2007 [13] dataset and focus on the "cat" and "dog" categories. Additional experimental results for other categories can be found in [48]. We train an independent classifier for each class. We follow the methodology outlined in [15] to construct $\Phi(\boldsymbol{x}, \boldsymbol{y})$. We crop image $\boldsymbol{x}$ to the bounding box $\boldsymbol{y}$, resize the resulting patch and pass it through a convolutional network pre-trained on a different dataset. We use here AlexNet [28] pre-trained on ImageNet [50] and take as $\Phi(\boldsymbol{x}, \boldsymbol{y})$ the output of the layer conv4. We use selective search to restrict $|\mathcal{Y}|$ to 1000 [66]. The max and top-$K$ oracles are implemented as exhaustive searches over this reduced set. We use $1 - \mathrm{IoU}$ as the task loss where $\mathrm{IoU}(\boldsymbol{y}, \boldsymbol{y}') = \mathrm{Area}(\boldsymbol{y} \cap \boldsymbol{y}')/\mathrm{Area}(\boldsymbol{y} \cup \boldsymbol{y}')$. Moreover, the ground truth label $\boldsymbol{y}$ is replaced by $\mathrm{argmax}_{\boldsymbol{y}' \in \mathcal{Y}} \mathrm{IoU}(\boldsymbol{y}, \boldsymbol{y}')$. We use the average precision (AP) as the performance metric for evaluation [13] .

**Methods.** The plots compare two non-smooth optimization methods, (a) SGD, which is a primal stochastic subgradient method with step-sizes chosen as $\gamma_t = \gamma_0/(1 + t/t_0)$, where $\gamma_0, t_0$ are parameters to be tuned, and returns the averaged iterate $\overline{\boldsymbol{w}}_t = 2/t(t+1) \sum_{j=1}^{t} j \boldsymbol{w}_j$ [29], and (b) BCFW, the Block-Coordinate Frank-Wolfe algorithm [30], with the tuning of the parameters proposed by the authors, and the averaged iterate as above (bcfw-wavg). The methods that use smoothing are SVRG [23] with constant smoothing and two variants of SC-SVRG, namely SC-SVRG-const, which uses constant smoothing (Scheme 1 in Table 2) and SC-SVRG-adapt, which uses adaptive smoothing (Scheme 2 in Table 2). Note that the step-size scheme of SGD does not follow from a classical theoretical analysis, yet performs better in practice than the one used by Pegasos [57].

**Parameters.** BCFW requires no tuning, while SGD requires the tuning of $\gamma_0$ and $t_0$. The SVRG-based methods require the tuning of a fixed learning rate. Moreover, SVRG and SC-SVRG-const also require tuning the amount of smoothing $\mu$. The validation $F_1$ score and the train loss are used as the tuning criteria for named entity recognition and visual object localization respectively. A fixed budget $T_{\mathrm{inner}} = n$ is used as the stopping criteria in Algorithm 1. This corresponds to the one-pass heuristic of [36], who found the theoretical stopping criteria to be overly pessimistic. We use the value $\kappa_k = \lambda$ for SC-SVRG-adapt. All smooth optimization methods turned out to be robust to the choice of $K$ for the top-$K$ oracle (Fig. 3) - we use $K = 5$ for named entity recognition and $K = 10$ for visual object localization.

**Experiments.** We present in Fig. 2 the convergence behavior of the different methods on the named entity recognition and visual object localization tasks. We plot the error on the training set vs. the number of oracle calls and the performance metric on a held-out set vs. the number of oracle calls.

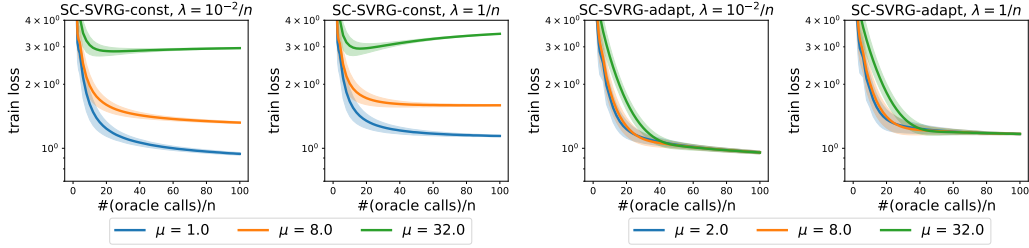

(a) Effect of smoothing hyperparameter on SC-SVRG-const and SC-SVRG-adapt for CoNLL-2003.

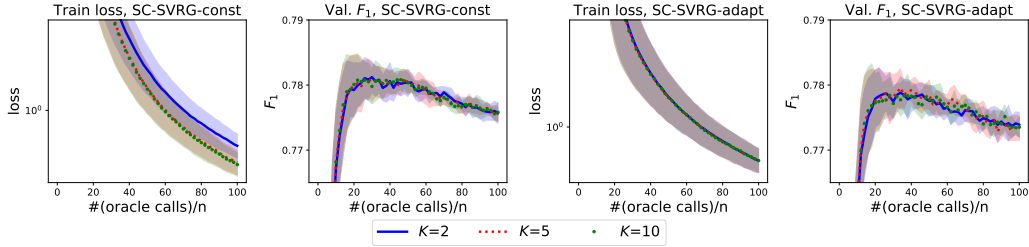

(b) Effect of hyperparameter $K$ on SC-SVRG-const and SC-SVRG-adapt for CoNLL-2003.

Figure 3: Effect of hyperparameters $\mu$ and $K$ on SC-SVRG-const and SC-SVRG-adapt.

As we can see in Fig. 2, the proposed methods converge faster in terms of training error while achieving a competitive performance in terms of the performance metric on a held-out set. Furthermore, BCFW and SGD make twice as many actual passes as SVRG based algorithms. In Fig. 3, we explore the effect of the parameters $\mu$ and $K$ on the convergence of the different methods. We can see that SC-SVRG-adapt is rather robust to the choice of $\mu$, while SC-SVRG-const and SVRG are more sensitive to the choice of $\mu$. Therefore SC-SVRG-adapt seems to appear as the most practical variant of our approach. We can also notice that SC-SVRG-adapt is rather robust to the choice of $K$. Setting $K = 5$ is sufficient here to obtain competitive results.

# 5 Related Work

The general framework for global training of structured prediction models was introduced in [4] and applied to handwriting recognition in [3] and to document processing in [5].

**Smooth inference oracles.** Smooth inference oracles with $\ell_2^2$-smoothing echo older heuristics in speech and language processing [25]. In the probabilistic graphical models literature, efficient algorithms to solve the top-$K$ inference combinatorial optimization problems were studied under the name "$M$-best MAP" in [54, 45, 68]. See [48] for a longer survey. Previous works considering smooth inference oracles yet encompassed by our framework can be found in [22, 24, 51, 41]. Instances of smooth inference oracles framed in the context of first-order optimization were studied in [58, 69] and in [38]. We framed here a general notion of smooth inference oracles in the context of first-order optimization. The framework not only includes previously proposed inference oracles but also introduces new ones.

Related ideas to ours appear in the independent works [39, 44]. These works partially overlap with ours, but the papers choose different perspectives, making them complementary to each other. In [39], the authors proceed differently when, *e.g.*, smoothing inference based on dynamic programming. Moreover, they do not establish complexity bounds for optimization algorithms making calls to the resulting smooth inference oracles. We define smooth inference oracles in the context of black-box first-order optimization and establish worst-case complexity bounds for incremental optimization algorithms making calls to these oracles. Indeed we relate the amount of smoothing controlled by $\mu$ to the resulting complexity of the optimization algorithms relying on smooth inference oracles.

**Batch and incremental optimization algorithms.** Several families of algorithms for structural support vector machines were proposed. Table 3 gives an overview with their oracle complexities. Early works [59, 65, 21, 62] considered batch dual quadratic optimization (QP) algorithms.

Table 3: Convergence rates given in the number of calls to various oracles for different optimization algorithms on the learning problem (1) in case of SSVMs (2). The rates are specified in terms of the target accuracy $\epsilon$, the number of training examples $n$, the regularization $\lambda$, the size of the label space $|\mathcal{Y}|$, the feature norm [48]. The rates are specified up to constants and factors logarithmic in the problem parameters. The dependence on the initial error is ignored. * denotes algorithms that make $O(1)$ oracle calls per iteration.

| Algo. (*exp* oracle) | # Oracle calls |
|---|---|
| Exponentiated gradient* [7] | $\dfrac{(n + \log|\mathcal{Y}|)R^2}{\lambda\epsilon}$ |
| Excessive gap reduction [69] | $nR\sqrt{\dfrac{\log|\mathcal{Y}|}{\lambda\epsilon}}$ |
| This work*, fixed smoothing, entropy smoother | $\sqrt{\dfrac{nR^2\log|\mathcal{Y}|}{\lambda\epsilon}}$ |
| This work*, adaptive smoothing, entropy smoother | $n + \dfrac{R^2\log|\mathcal{Y}|}{\lambda\epsilon}$ |

| Algo. (*max* oracle) | # Oracle calls |
|---|---|
| BMRM [62] | $\dfrac{nR^2}{\lambda\epsilon}$ |
| QP 1-slack [21] | $\dfrac{nR^2}{\lambda\epsilon}$ |
| Stochastic subgradient* [57] | $\dfrac{R^2}{\lambda\epsilon}$ |
| Block-Coordinate Frank-Wolfe* [30] | $n + \dfrac{R^2}{\lambda\epsilon}$ |

| Algo. (*top-K* oracle) | # Oracle calls |
|---|---|
| This work*, fixed smoothing, $\ell_2^2$ smoother | $\sqrt{\dfrac{n\widetilde{R^2}}{\lambda\epsilon}}$ |
| This work*, adaptive smoothing, $\ell_2^2$ smoother | $n + \dfrac{\widetilde{R}^2}{\lambda\epsilon}$ |

The stochastic subgradient method considered by [49, 57] operated directly on the non-smooth primal formulation [49, 57]. More recently, [30] proposed a block coordinate Frank-Wolfe (BCFW) algorithm to optimize the dual formulation of structural support vector machines; see also [46] for variants and extensions. Saddle-point or primal-dual optimization algorithms are another family of algorithms, including the dual extra-gradient algorithm of [61] and the mirror-prox algorithms of [8, 19]. In [47], an incremental optimization algorithm for saddle-point problems is proposed. However it is unclear how to extend it to the structured prediction problems we consider here. Incremental optimization algorithms for conditional random fields were proposed in [52]. We focus here on primal optimization algorithms in order to be able to train structured prediction models with affine or nonlinear mappings with a unified approach, and on incremental optimization algorithms in order to be able to scale to large datasets.

# 6    Conclusion

We introduced a general notion of smooth inference oracles in the context of black-box first-order optimization. This allows us to set the scene to extend the scope of fast incremental optimization algorithms to structured prediction problems owing to a careful blend of a smoothing strategy and an acceleration scheme. We illustrated the potential of our framework by proposing a new incremental optimization algorithm to train structural support vector machines both enjoying worst-case complexity bounds and demonstrating competitive performance on two real-world problems. This work paves the way to faster incremental primal optimization algorithms for deep structured prediction models explored in more detail in [48]. There are several potential venues for future work. When there is no discrete structure that admits efficient inference algorithms, it could be beneficial to not treat inference as a blackbox numerical procedure [40, 16, 17]. Instance-level improved algorithms along the lines of [17] could also be interesting to explore.

**Acknowledgements**    This work was supported by NSF Award CCF-1740551, the Washington Research Foundation for innovation in Data-intensive Discovery, and the program "Learning in Machines and Brains" of CIFAR.

## Footnotes

[1]We say $f$ is $L$-smooth with respect to $\|\cdot\|$ when $\nabla f$ exists everywhere and is $L$-Lipschitz with respect to $\|\cdot\|$. Smoothness and strong convexity are taken to be with respect to $\|\cdot\|_2$ unless stated otherwise.

[2] $\|\boldsymbol{A}\|_{\beta,\alpha} := \max\{\boldsymbol{u}^\top \boldsymbol{A} \boldsymbol{w} \mid \|\boldsymbol{u}\|_\alpha \leq 1, \|\boldsymbol{w}\|_\beta \leq 1\}.$

[3] The notation $O(\cdot)$ may hide constants and factors logarithmic in problem parameters. See [48] for detailed complexities.

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
