[Reviews · NeurIPS 2018]

Reviewer 1



The paper explores how smooth maximization oracles can be used to optimize structured support vector machines with recent first-order incremental optimization methods which have fast convergence rates. The paper explores the notion of an l2-smoothed and an entropy-smoothed maximization oracle. The appendix explains how to implement an l2-smoothed maximization oracle in practice with a top-k oracle, which seems necessary to implement and understand the algorithms in the paper. While simply coupling this smoothing oracle with existing algorithms such as SVRG can provide good results the paper also explores Nesterov-style accelerated optimization methods, in the catalyst framework. These have the advantage that it's straightforward to anneal the amount of smoothing during training to better approximate the non-smoothed loss. I did not carefully check all the theorems and convergence rates, but the basic derivations of the algorithms is clear and easy to follow. The experimental results are somewhat convincing, but I'd like to see more thorough baselines. For example, it's possible to change the learning rate of Pegasos and similar algorithms in ways that do not correspond to the original paper but get better performance. Similarly, it'd be interesting to see more results on how methods such as SVRG perform when no smoothing is added, to understand the value of smoothing. That said, the idea of smoothing for structured prediction is interesting, and this paper leaves many avenues open for future work, so I think this belongs in NIPS. After reading the author response my score is unchanged; I still think this paper belongs in NIPS. I encourage the authors to add the extra experiments in the response to the paper to clarify better what aspects of the methods are important.

Reviewer 2



Overview: This paper proposes an accelerated variance-reduction algorithm for training structured predictors. In this approach the training objective is augmented with a proximal term anchored with a momentum point (eq (3)), the loss is smoothed using Nesterov's smoothing method (adding entropy or L2 to the dual), and a linear-rate solver (SVRG) is applied to the resulting objective in the inner loop. This achieves accelerated convergence rates for training. Comments: * I think that the connection to structured prediction is somewhat weak. In particular, the analysis uses the finite sum and smoothability of the training objective. The first is not specific to structured prediction, and the latter has been studied extensively in previous work, including for learning (see next point). Inference is assumed to be an atomic operation performed via calls to an inference oracle -- assuming access to efficient top-k and expectations computation, and there is no leveraging of instance-level structure. * I don't think it's fair to say that smooth inferene oracles are a contribution of this paper (line 40). Entropic smoothing was used much earlier, for example: - Convex Relaxation Methods for Graphical Models: Lagrangian and Maximum Entropy Approaches, Johnson 2008. - Fast and smooth: Accelerated dual decomposition for MAP inference, Jojic et al. 2010. - [Adaptive smoothing] Efficient MRF Energy Minimization via Adaptive Diminishing Smoothing, Savchynskyy et al. 2012. Also in the context of learning (SSVMs), e.g.: - Blending Learning and Inference in Conditional Random Fields; Hazan et al. JMLR 2016. L2 smoothing for inference was used more recently, for example: - Smooth and Strong: MAP Inference with Linear Convergence, Meshi et al. 2015. - [For deep structured models] From Softmax to Sparsemax: A Sparse Model of Attention and Multi-Label Classification, Martins and Astudillo 2016. * Top-K oracles may sometimes be more expensive than max oracles (also true for marginal oracles). It’s worth providing more details on that to clarify when it’s not a good idea to use this type of smoothing. * In the experiments (Fig 2), the accelerated methods don't seem to have an advantage over plain SVRG in terms of test performance. Minor comments: * Lines 62-63: the remark about the loss being decomposable is not really necessary since nothing is assumed about the score (line 58). Instead, perhaps better to directly extend the assumption on the *oracle* to include (2). Otherwise, score decomposition may be mentioned. * Line 81: it would be good to clarify that u_2 depends on mu via notation, e.g., u_2(y;\mu) or u_\mu(y). * Line 194 [typo]: 'set reduced set'. ================= After author response: To better place this work in context of existing literature on structured prediction, I suggest you down-tone the contribution on smoothing of inference and cite a few works on Nesterov smoothing of inference objectives. I also think you should add a discussion on complexity of the required oracles making it clear that smoothing often requires more expensive oracles.

Reviewer 3



The main message of this paper is to introduce a smooth inference for structured prediction, which practically is not novel as other methods such as gradient descent inference use a continuous output variables as a relation of discrete variables and smoothen the inference using L2 or using entropy, and projects the output variables into simplex of probabilities after each steps of gradient descent. However, this work presents the idea in a carefully described framework with theoretical guarantees that was missing from previous works. The next contribution of the paper is to use the smooth SVRG as the M in the catalyst framework. My main problem with the paper is that I am not sure why these two parts are presented in a single paper since each targeted different aspect of the structured prediction problem (the first one is smoothing inference, and the second one targeted smoothing the weight optimization by introducing well-conditioned convex problem). I guess the message of these two parts is very different if not contradictory. As I expect and confirmed by Figures 8-10, I can conclude that smoothing inference actually helps test set generalization rather than having a better rate for train loss convergence (comparing to BCFG that mostly has better rates but does not generalize well to the test data), which is not very surprising since the inference returns a combination of output structures instead the most likely one.